# Optimising Access Surgery in Senior Haemodialysis Patients (OASIS): study protocol for a multicentre randomised controlled trial

Boudewijn DC Heggen ,[1] Chava L Ramspek ,[2] Koen E A van der Bogt ,[3,4] Michiel W de Haan,[5] Marc H Hemmelder,[6] Mickaël J C Hiligsmann ,[7] Magda M van Loon,[1] Joris I Rotmans,[8] Jan H M Tordoir,[1] Friedo W Dekker,[2] Geert Willem H Schurink,[1] Maarten G J Snoeijs,[1] on behalf of the OASIS study group

For numbered affiliations see end of article.

**Correspondence to**
Mr Boudewijn DC Heggen;
boudewijn.heggen@mumc.nl

## ABSTRACT

**Introduction** Current evidence on vascular access strategies for haemodialysis patients is based on observational studies that are at high risk of selection bias. For elderly patients, autologous arteriovenous fistulas that are typically created in usual care may not be the best option because a significant proportion of fistulas either fail to mature or remain unused. In addition, long-term complications associated with arteriovenous grafts and central venous catheters may be less relevant when considering the limited life expectancy of these patients. Therefore, we designed the Optimising Access Surgery in Senior Haemodialysis Patients (OASIS) trial to determine the best strategy for vascular access creation in elderly haemodialysis patients.

**Methods and analysis** OASIS is a multicentre randomised controlled trial with an equal participant allocation in three treatment arms. Patients aged 70 years or older who are expected to initiate haemodialysis treatment in the next 6 months or who have started haemodialysis urgently with a catheter will be enrolled. To detect and exclude patients with an unusually long life expectancy, we will use a previously published mortality prediction model after external validation. Participants allocated to the usual care arm will be treated according to current guidelines on vascular access creation and will undergo fistula creation. Participants allocated to one of the two intervention arms will undergo graft placement or catheter insertion. The primary outcome is the number of access-related interventions required for each patient-year of haemodialysis treatment. We will enrol 195 patients to have sufficient statistical power to detect an absolute decrease of 0.80 interventions per year.

**Ethics and dissemination** Because of clinical equipoise, we believe it is justified to randomly allocate elderly patients to the different vascular access strategies. The study was approved by an accredited medical ethics review committee. The results will be disseminated through peer-reviewed publications and will be implemented in clinical practice guidelines.

**Trial registration number** NL7933.

**Protocol version and date** V.5, 25 February 2021.

### Strengths and limitations of this study

► This is the first prospective trial randomly allocating elderly haemodialysis patients to the three commonly used vascular access strategies.
► By enrolling patients on referral for vascular access creation, the trial will reflect everyday practice and study patients with predialysis vascular access creation as well as patients who have started dialysis urgently with a temporary catheter.
► The primary outcome, that is, the number of access-related interventions required for each year of haemodialysis treatment, corresponds to the recently proposed core outcome measure for clinical trials in vascular access.
► A previously published mortality prediction score was validated to determine a cut-off point for exclusion of elderly patients with an unusually long life expectancy for whom a fistula is still considered to be the best vascular access.

## INTRODUCTION

The haemodialysis population has been growing older and older over the past decades, and the median age of patients starting haemodialysis treatment today is 66 years.[1] These patients need a reliable vascular access that may be provided by autologous arteriovenous fistulas, prosthetic arteriovenous grafts and central venous catheters. Autologous fistulas are created by surgically anastomosing a superficial vein to an artery in the upper extremity. After creation of the fistula, vascular remodelling will increase venous outflow diameter and blood flow, a process referred to as maturation. Functional fistulas are generally associated with the best long-term outcomes. However, a substantial proportion of fistulas fail to mature and never

become functional.[2] Arteriovenous grafts are placed by interposing a prosthetic vascular graft between an artery and a vein in a subcutaneous track in the upper extremity. Grafts do not need time to mature but are associated with a higher incidence of infection, stenosis and thrombosis than functional fistulas.[3] Central venous catheters are inserted in the internal jugular vein with a minimally invasive procedure and can be used immediately after placement. However, catheters have been associated with increased bloodstream infections and mortality compared with other vascular access types.[3 4] Studies comparing these different vascular access strategies typically focus on functional outcomes, and little is known about differences in patient-reported outcome measures (PROMs) and healthcare costs.

Clinical practice guidelines on vascular access have long recommended the creation of autologous arteriovenous fistulas regardless of patient characteristics.[5] However, this recommendation is based solely on observational studies that are at high risk of bias. Patients who have a fistula created prior to starting dialysis generally have more favourable baseline characteristics and a lower mortality risk than other patients.[6] These favourable patient characteristics may account for two-thirds of the survival benefit seen in patients using arteriovenous fistulas for haemodialysis treatment instead of central venous catheters.[7] Further selection bias will occur as patients are analysed according to vascular access use. This tends to favour autologous fistulas since the poor subset of patients with fistulas that have failed to mature will go on to use a different type of vascular access. When primary fistula failures are taken into account, the cumulative patency of fistulas and grafts become comparable.[8] Finally, observational studies based on dialysis registries do not include patients with arteriovenous fistulas created before dialysis initiation that were never used because of patient death or stable renal function.

Uncertainty regarding the best type of vascular access is particularly important for elderly patients starting haemodialysis treatment. In these patients, autologous arteriovenous fistulas have a 40% increased risk of non-maturation and early failure.[2] Moreover, 30% of fistulas created before initiation of dialysis treatment in elderly patients will never be used because of patient death, stabilisation of renal function or choice for conservative treatment as the patient's condition worsens over time.[9] Furthermore, the increase in long-term complications associated with arteriovenous grafts may be less relevant for elderly patients with a limited life expectancy. Finally, elderly patients have a 30% lower risk of complications of central venous catheters, including bloodstream infections.[4 10] Considering this clinical equipoise, the need for better evidence on vascular access strategies in elderly haemodialysis patients is broadly recognised. Indeed, international experts on vascular access surgery as well as patient representatives have called for a randomised trial comparing vascular access strategies in elderly haemodialysis patients.[5 11]

We performed a comprehensive search in international trial registers for ongoing and planned randomised trials on vascular access creation in elderly haemodialysis patients. Four ongoing trials were identified that allocate participants to arteriovenous fistula or graft surgery. One trial has a single-centre design and plans to enrol 270 haemodialysis patients aged 70 years or older. The participants are followed up for 2 years after vascular access creation for the primary outcomes, which are vascular access use as well as patency.[12] The remaining three trials are pilot studies evaluating the feasibility of the trial design. These pilot studies allocate patients aged 65 years or older, either at the time of referral for vascular access, at dialysis initiation or after dialysis initiation with a catheter.[13–15] We also identified two trials allocating participants to fistula creation or catheter insertion. The ACCESS HD feasibility study allocates patients aged 55 years or older who have already commenced haemodialysis treatment with a catheter to either fistula creation or continued catheter use.[11] The AUSTrian Randomized Interventional Study on Dialysis Accesses (AUSTRIA) aims to allocate 220 participants aged 60 years or older to one of the two treatment arms, with access-related complications being the primary outcome measure.[16] Although these ongoing studies will surely provide valuable information, none of the trials cover the full spectrum of vascular access strategies in elderly haemodialysis patients.

The aim of the Optimising Access Surgery in Senior Haemodialysis Patients (OASIS) trial is to determine the best vascular access strategy for elderly haemodialysis patients. We hypothesise that arteriovenous grafts and central venous catheters are superior to autologous arteriovenous fistula for elderly patients starting haemodialysis treatment and result in reduced access-related intervention rates, greater patient satisfaction and lower healthcare costs.

## METHODS AND ANALYSIS
### Study design
OASIS is a parallel group, multicentre, randomised controlled trial with a superiority framework and a 1:1:1 individual participant treatment allocation ratio over three study arms. Given the nature of the interventions, blinding healthcare providers, participants or outcome assessors for treatment assignment are not possible. However, data analysis will be performed without knowledge of treatment assignment by coding the treatment arms and unlocking the code only after statistical analysis is completed. Some 17 hospitals, together treating approximately 50% of all haemodialysis patients in the Netherlands, are expected to participate.

### Study population
On referral for permanent vascular access creation, all patients will have a preoperative duplex ultrasound assessment of the blood vessels of the upper extremities. This

preoperative workup may be supplemented by central venous phlebography and arterial imaging as considered appropriate by the local vascular access team. All potential participants will receive oral and written information about the trial by the local investigators and are registered in a screening log.

## Inclusion criteria

► Age 70 years or older.
► End-stage renal disease with unlikely recovery of kidney function according to the attending nephrologist.
► Haemodialysis is the intended long-term modality of treatment for end-stage renal disease.
► Fit for any of the proposed vascular access interventions as determined by the local multidisciplinary vascular access team.
► (1) Expected to start haemodialysis treatment within 6 months at the time of treatment assignment, or (2) treated with haemodialysis for 6 months or less at the time of treatment assignment using a tunnelled or non-tunnelled central venous catheter for vascular access.
► Planning to remain in one of the participating dialysis centres for at least 1 year.
► Suitable vascular anatomy for all types of vascular access based on duplex ultrasound of the arms, defined as[5]
  At least one suitable configuration for an arteriovenous fistula using minimal arterial and venous diameters of 2 mm for radiocephalic fistulas and 3 mm for brachiocephalic and brachiobasilic fistulas;
  At least one suitable configuration for an arteriovenous graft using minimal arterial and venous diameters of 3 and 4 mm, respectively
  At least one open internal jugular vein for a central venous catheter.

## Exclusion criteria

► Patent arteriovenous fistula or graft already in place.
► Prior unsuccessful arteriovenous fistula or graft vascular access surgery.
► Kidney transplantation scheduled within 6 months.
► Metastatic malignancies or other condition associated with a life expectancy of <6 months, in the opinion of the attending nephrologist.
► Unable to provide informed consent.
► Dusseux risk score of ≤4, indicating an unusually long life expectancy.[17]

Patients will be enrolled by local investigators if they meet the eligibility criteria and provide informed consent. Local investigators will be informed of treatment assignment through a web-based service, after which vascular access creation is planned according to usual clinical care for the allocated vascular access type at the study site (figure 1). The treatment allocation sequence is generated by an independent data management centre using a computer-based random number-producing algorithm. Block randomisation is implemented with randomly

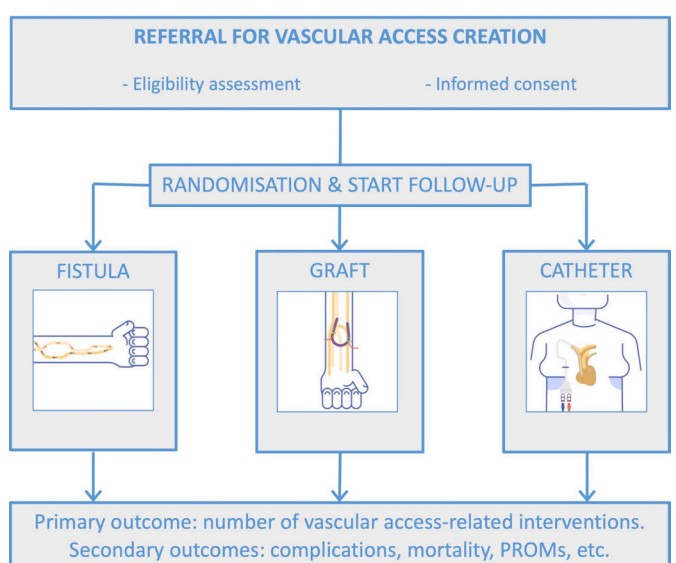

**Figure 1** Study flowchart. On referral for vascular access creation, patients are screened for eligibility. Patients who meet the inclusion and exclusion criteria and provide informed consent will be randomised and allocated to one of the treatment strategies. Follow-up for the primary and secondary outcomes starts at the moment of treatment allocation. PROM, patient-reported outcome measure.

varying block sizes of 3 and 6, and randomisation will be stratified by treatment centre. Follow-up starts immediately after treatment allocation.

## External validation of a mortality prediction model

Elderly haemodialysis patients form a heterogeneous population in terms of comorbidities and life expectancy. The majority of elderly haemodialysis patients have multiple comorbidities and a relatively short life expectancy, making the long-term performance of the vascular access less relevant. However, for the small group of elderly patients with an unusually long life expectancy, autologous arteriovenous fistulas are considered to be the best vascular access, given their low complication rates after maturation. To identify this subset of elderly patients that should be excluded from randomisation, we performed an external validation of the mortality prediction score developed by Dusseux *et al*.[17] This prediction score contains items readily identified from the medical history and can be completed before initiation of haemodialysis treatment.

The mortality prediction score was validated in the Netherlands Cooperative Study on the Adequacy of Dialysis (NECOSAD).[18 19] In this multicentre observational cohort study, adult patients starting dialysis treatment between 1997 and 2007 in the Netherlands were prospectively followed up until 1 February 2015. For the current validation, we restricted the sample to haemodialysis patients aged 70 years and older. This resulted in 494 patients, 411 of whom died during follow-up with a median survival of 2.3 years (IQR 1.1–4.4 years). Overall, baseline characteristics in the validation cohort were comparable

**Table 1** Baseline characteristics of the NECOSAD validation cohort compared with the Dusseux derivation cohort[17]

| | NECOSAD cohort n=494 | Dusseux cohort n=8955 |
|---|---|---|
| Gender (male) (%) | 61 | 60 |
| Age (years) | 76 (73–79) | 78 (74–82) |
| Diabetes (%) | 25 | 38 |
| Ischaemic heart disease (%)* | 31 | 14 |
| Peripheral vascular disease (%) | 24 | 27 |
| Cerebrovascular disease (%) | 14 | 12 |
| Congestive heart failure (%) | 19 | 34 |
| Dysrhythmia (%)† | 22 | 25 |
| Respiratory disease (%)‡ | 12 | 13 |
| Active malignancy (%) | 4 | 11 |
| Psychiatric disorder (%)§ | 3 | 4 |
| Mobility (%)¶** | | |
| Walks without help | 64 | 69 |
| Needs assistance for transfers | 32 | 22 |
| Totally dependent for transfers | 4 | 9 |
| BMI (kg/m$^2$) (median, %)†† | | |
| <21 | 13 | 18 |
| 22–25 | 43 | 35 |
| 25–30 | 33 | 33 |
| >30 | 10 | 14 |
| Central Venous Catheter at dialysis initiation (%) | 23 | 42 |

*Ischaemic heart disease was defined as a history of angina pectoris or myocardial infarction.
†Vitamin K antagonist use was used as a proxy for cardiac dysrhythmia.
‡Respiratory disease was defined as the need to take pulmonary medication on a daily basis.
§Psychiatric disorder was defined as dementia, depression or other psychiatric disease.
¶The Karnofsky score was used as a proxy for mobility, where a score of >60 was considered as 'walks without help', a score of 50–60 as 'needs assistance for transfers' and a score of <50 as 'totally dependent for transfers'.
**32% of the data concerning mobility was missing in the Dusseux cohort.
††29% of the data concerning BMI was missing in the Dusseux cohort.
BMI, body mass index; NECOSAD, Netherlands Cooperative Study on the Adequacy of Dialysis.

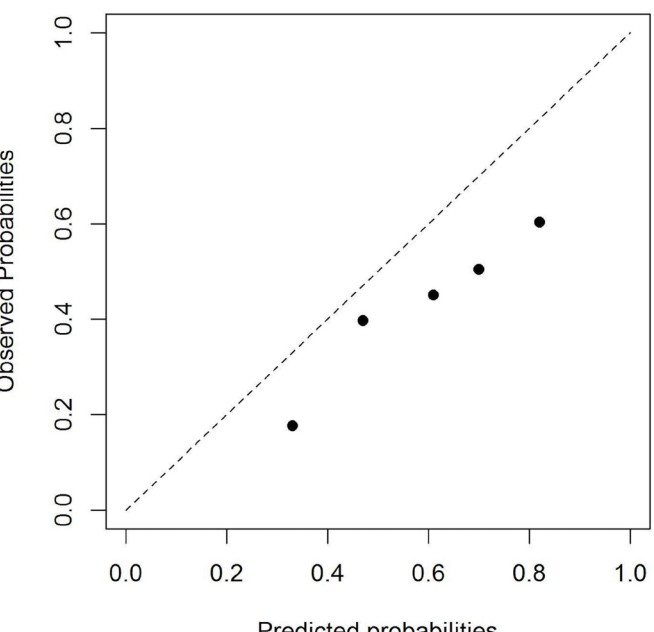

**calibration plot dusseux score**

**Figure 2** Calibration plot presenting the mortality risk as predicted by the Dusseux risk score and the observed risk in the Netherlands Cooperative Study on the Adequacy of Dialysis cohort, such that the 45° line indicates perfect agreement between predicted and observed risks.

to those in the Dusseux cohort (table 1).[17] Missing data were assumed to be largely missing at random. Therefore, 10-fold multiple imputation with fully conditional specification was performed using the R package 'mice'. All variables of the mortality prediction score and the outcome were included in the imputation model.[20 21]

Each individual's risk score was calculated with the original point system as presented by Dusseux (online supplemental data 1). Risk scores ranged from 0 to 28, with a higher score corresponding to a higher risk of mortality. From the individual risk scores, the discrimination for mortality within 2 years was assessed by c-statistics that were pooled over the 10 imputation datasets according to Rubin's rules.[22] The Dusseux mortality risk score had a c-statistic of 0.68 (95% CI 0.63 to 0.73) in the NECOSAD validation cohort. Furthermore, the cohort was divided into five groups with increasing mortality risk. A calibration plot of predicted versus observed mortality risk was computed. Calibration was moderate as the model tended to overpredict the mortality risk (53% average predicted risk and 38% average observed risk, figure 2). Finally, we determined a sensible cut-off point for identifying a subset of patients with an unusually long life expectancy. A cut-off between a score of ≤4 and ≥5 was chosen to separate patients with unusually good outcomes (consisting of 22% of patients in the validation cohort with a median survival of 5.0 years, 95% CI 4.3 to 5.4) from elderly haemodialysis patients with a poor life expectancy (consisting of the remaining 78% of patients in the validation cohort with a median survival of 2.2 years, 95% CI 1.9 to 2.5; figure 3).

### Interventions

The surgical strategies in the usual care and intervention groups are part of standard clinical care. Because we aim to compare the different surgical strategies for vascular access creation in everyday practice, we will not interfere with surgical protocols at the study sites. However, we

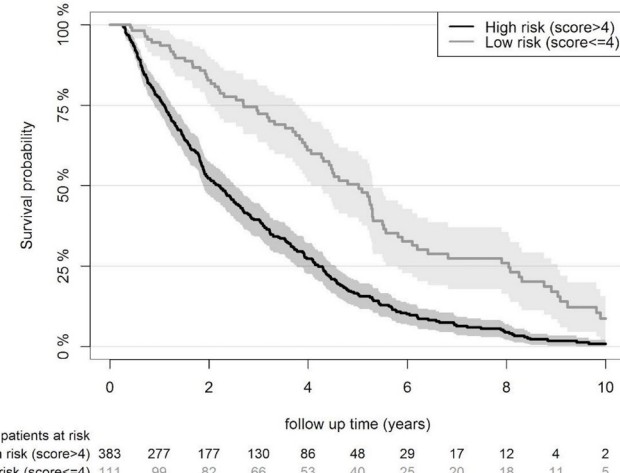

**Figure 3** Kaplan-Meier curve comparing the survival probability for the high risk and the low risk groupS in the Netherlands Cooperative Study on the Adequacy of Dialysis cohort.

expect the study sites to follow current best practice and guideline recommendations.[5]

### Autologous arteriovenous fistulas

Fistulas are preferably created at the most distal site with adequate blood vessels, in the non-dominant arm. It is recommended to create the fistula 3–6 months before the expected start of haemodialysis treatment. In case of fistula failure, a new arteriovenous fistula may be created at the next available site.

### Prosthetic arteriovenous grafts

Patients who are allocated to the graft strategy will have a commercially available prosthetic vascular graft implanted for haemodialysis access. It is recommended to place the arteriovenous graft 2 weeks before the expected start of haemodialysis treatment with antibiotic prophylaxis. Placement of an early-cannulation graft is recommended for patients who require a more urgent start of haemodialysis to avoid the use of a temporary central venous catheter. In case of graft failure despite thrombectomy, a new graft may be placed.

### Central venous catheters

Patients who are allocated to the catheter strategy will have a dialysis catheter inserted, unless they have already started haemodialysis with a tunnelled catheter. We recommend placing a tunnelled catheter just before the start of haemodialysis treatment. We advise following published guidelines for the prevention and treatment of catheter infections.[23] In case of catheter dysfunction despite local thrombolytic agents, the catheter can be exchanged.[24]

For the arteriovenous fistula and graft group, surveillance and pre-emptive intervention for stenosis will take place according to usual practice at the study sites. In case of failure, both endovascular and surgical interventions can be undertaken to restore patency. If the assigned vascular access strategy proves not to be feasible, patients will cross over to one of the other treatment groups after discussion of the case with the principal investigator. Use of a temporary central venous catheter is not considered as a treatment group crossover.

### Primary study outcome

The primary study outcome is the number of access-related interventions required for each person-year of haemodialysis treatment. To analyse this, we will divide the total number of interventions during follow-up by the total number of haemodialysis years. The primary outcome corresponds to the proposed core outcome measure for haemodialysis vascular access.[25] This outcome measure includes all percutaneous access interventions, surgical access procedures and catheter interventions from randomisation until the end of the study period or death. The outcome measure specifically includes interventions before dialysis initiation and after dialysis cessation in the occasional patient who stops haemodialysis treatment after kidney transplantation, peritoneal dialysis, recovery of renal function or refusal of further haemodialysis. These vascular access interventions before dialysis initiation and after dialysis cessation are included as they are part of the standard care that this trial evaluates. Patients who discontinue haemodialysis treatment will therefore remain in the trial. Access-related complications that are resolved using conservative or pharmacological treatment are not counted as interventions.

### Secondary study outcomes
#### Complications

Access-related complications requiring pharmacological, surgical or endovascular interventions will be registered from randomisation until the end of the study period or death. Furthermore, the number of days admitted to hospital or visiting outpatient clinics for any reason and for vascular access-related reasons per person-year will be registered. Finally, all-cause mortality from randomisation to the end of the study period will be registered.

#### Patient-reported outcome measures

► Short-Form Health Survey (SF-12) and the Dialysis Symptom Index (DSI) measure generic health-related quality of life as well as disease-specific symptom burden.[26 27]
► The Short-Form Vascular Access Questionnaire (SF-VAQ) measures haemodialysis patients' satisfaction with their vascular access.[28]
► The 5-Level EuroQol 5-Dimensional Questionnaire (EQ-5D-5L) measures health-state utility values.[29]

The SF-VAQ will be administered on a monthly basis, whereas the SF-12, DSI and EQ-5D-5L will be administered at 3-month intervals. To lower their burden, participants who have started haemodialysis treatment will receive the questionnaires during these treatments and will get assistance from a nurse. The questionnaires will be administered from randomisation until the 1 year after study or dialysis initiation, whichever is later.

Other outcome measures will be registered for exploratory analyses and are defined according to the ESVS guidelines on vascular access[5]:

► Primary, assisted primary, secondary and functional patency of vascular access.
► Time until vascular access maturation (definition according to the American Society of Nephrology).[30]
► Time until functional vascular access.
► The number of haemodialysis sessions with inadequate haemodialysis dose (Kt/V) and with cannulation difficulties or failure. Cannulation difficulties or failures will be recorded with the use of monthly questionnaires asking dialysis nurses whether more than one cannulation attempt was required, whether two-needle cannulation was impossible or if cannulation was painful.

## Quality and safety

Data are entered on electronic case report forms by trained investigators using numerical codes that cannot be traced to the individual subjects. The subject identification code list is kept by the principal investigator at each study site. Any serious adverse events that may occur will be reported to the principal investigator without undue delay after obtaining knowledge of the events. To evaluate differences in serious adverse events and mortality between treatment arms, an interim safety analysis will be performed and presented to a data safety monitoring board every 6 months. Significant differences on safety outcomes may lead to discontinuation of the trial. Furthermore, an independent monitor will be appointed to assess trial processes and data.

## Sample size calculation

Sample size was estimated for the primary study outcome of access-related intervention rates. Since two intervention groups will be compared with the usual care group, we used Bonferroni correction to account for multiple comparisons, which resulted in an alpha of 0.025. The average number of interventions per year for patients receiving an autologous arteriovenous fistula has previously been reported at 2.48 interventions per year.[31] The clinically relevant effect size was considered to be one percutaneous intervention per year and 0.5 surgical interventions per year. Since these interventions occur at a ratio of 1.0:0.7,[31] a reduction of the overall intervention rate by 0.80 interventions per year was used as the clinically relevant effect for sample size calculations.

With these assumptions, a total number of 195 patients (65 patients in each treatment arm) with 1 year of follow-up achieves 80% power to detect a 0.80 decrease in the number of access-related interventions per person-year of haemodialysis treatment between the intervention groups and the usual care group using a two-sided, large-sample z-test of the Poisson event–rate difference at a significance level of 0.025.[32]

Since the study includes patients with a limited life expectancy, a substantial proportion of participants are expected to die before contributing 1 year of follow-up. To compensate for the resulting loss of statistical power, the trial has a variable follow-up with trial closeout for all patients at the time when the last patient enrolled has 1 year of follow-up. Since participant recruitment is expected to take 2 years, the resulting additional follow-up time will more than compensate for the loss of follow-up due to patient mortality in the first year of the study period.

## Data analysis plan

The access-related intervention rates will be analysed using a general linear model with Poisson distribution and identity link, and with time as off-set variable. Both intervention (ie, the prosthetic arteriovenous graft and central venous catheter) groups will be compared with the usual care (autologous arteriovenous fistula) group. Non-adherence to the allocated treatment group is expected as patients cross over to another surgical strategy for vascular access when the randomised strategy is no longer feasible. The primary analysis will be on the intention-to-treat population. Exploratory on-treatment analyses will be performed as well. A subgroup analysis will be performed for patients who had already started haemodialysis using a catheter before inclusion.

Formal statistical comparisons will be made for the following secondary study outcomes:

► SF-12/DSI and SF-VAQ summary scores using generalised estimating equations.
► The number of access-related serious adverse events per person-year using general linear models with Poisson distribution and mortality using Cox regression.
► The number of days admitted to the hospital or visiting outpatient clinics for any reason per person-year using Student's t-tests.

Primary outcome data are not expected to be missing, as interventions on vascular access will be reported in the patients' medical files. We expect little to no loss to follow-up since study participants will be observed three times per week in the dialysis unit and the trial requires no additional study visits. PROMs will be analysed using generalised estimating equations that allow for missing data. Other missing data will be handled by using 10 imputation cycles with regression methods, performing standard analyses for each imputation cycle and considering the variability across the imputation cycles in the final analysis.

## Economic analysis

A cost effectiveness analysis and a budget impact analysis comparing the different treatment strategies will be performed by collecting financial data from randomisation until the 1 year after study or dialysis initiation, whichever is later. The economic analyses will be done according to Dutch national guidelines and have a time horizon of 2 years that corresponds to the median life expectancy of the trial participants.[33]

For the cost effectiveness analysis, we will consider both healthcare sector costs and costs for patients and their families. Due to the age of our study population, absenteeism in paid work and production losses in the domestic sphere will not be considered. The relevant cost factors for the healthcare sector will be derived from hospital and pharmacy registration systems at the individual participant level. Costs to patients and families will be measured using a study-specific adaptation of the Medical Consumption Questionnaire developed by the Institute for Medical Technology Assessment at 3 month intervals.[34] After measurement of the relevant cost factors, monetary values will be assigned.

The budget impact analysis will be performed to estimate the financial consequences for the Dutch national healthcare budget for the different vascular access strategies in elderly haemodialysis patients. The perspectives that will be included in the analysis are the wider societal perspective including productivity losses, the narrower perspective of the public, and the perspective of healthcare providers and health insurance companies. Several scenarios will be included to assess the impact of different reimbursement options.

### Patient and public involvement

The Dutch Renal Patients Society (NVN) was involved in the design of OASIS. We consulted patient representatives on the feasibility of randomisation into the three treatment arms and on the choice of the primary outcome measure. Patient representatives also advised on the frequency and duration of patient-reported outcome measurements and approved the patient information letter.

## ETHICS AND DISSEMINATION
### Ethics
The study will be conducted according to the principles of the Declaration of Helsinki (Brazil, 2013) and the Medical Research Involving Human Subjects Act. The protocol was approved by an accredited medical ethics review committee in November 2019. Some physicians may consider it unethical to use central venous catheters in haemodialysis patients who are candidates for autologous arteriovenous fistula creation, considering the increased mortality and infection rate associated with catheter use. However, there is serious doubt whether a causal relationship between catheter use and mortality exists. The association between catheter use and mortality is derived from observational studies that are profoundly influenced by selection bias. Two examples of this are found in recent studies demonstrating that patient selection for fistula placement, even when they are dialysed with a catheter instead, explains at least two-thirds of the mortality benefit observed in patients with a fistula.[6 7] Furthermore, the majority of the excess mortality observed in patients who use a catheter is felt to be attributable to catheter-related bloodstream infections. However, during the run-in period of a multicentre randomised controlled trial, the incidence of catheter-related bloodstream

infections was only 1.0 per 1000 catheter days.[35] In elderly patients, the risk of infection is even lower than in the general haemodialysis population.[9] In a large cohort of dialysis patients with catheter-related bloodstream infections, only 1% of these infections were fatal.[36] Therefore, the observed survival benefit with fistulas can impossibly be explained by avoidance of access-related infections. Moreover, there is clinical equipoise in elderly haemodialysis patients because the balance between risks and benefits of the different types of vascular access is different in this population. Any increase in access-related complications associated with long-term use of grafts and catheters may be less relevant for elderly patients with an expected median survival of 2.2 years. Finally, our study has been designed to minimise the potential risk for participants by scheduling frequent safety analyses, by allowing treatment crossover in case of access complications and by excluding elderly patients with an unusually long life expectancy.

### Dissemination
The OASIS trial is expected to lead to new insights in vascular access surgery for elderly haemodialysis patients. After completion of the trial, the results will be disseminated among vascular surgeons, nephrologists, interventional radiologists and dialysis nurses through international peer-reviewed publications in medical journals and presentations at scientific meetings.

After the trial results have been published, its findings should be implemented into clinical practice. To identify possible facilitators and obstacles for implementation, OASIS will include a process evaluation. We will register the reasons for patients not participating in the randomised trial in a screening log, and we will study ineligible elderly haemodialysis patients in a parallel observational cohort study. Patient characteristics and outcomes of the randomised trial and the observational cohort will be compared to evaluate the generalisability of the trial. Furthermore, we will register study protocol violations during the enrolment and follow-up phase, and we will investigate any protocol violations, such as non-adherence to the assigned treatment, by interviewing the treating physician. This process evaluation will refine our implementation plan with strategies tailored to the specific clinical circumstances.

**Author affiliations**
[1]Department of Vascular Surgery, Maastricht University Medical Centre+, Maastricht, Netherlands
[2]Department of Clinical Epidemiology, Leiden University Medical Centre, Leiden, Netherlands
[3]Department of Surgery, Haaglanden Medical Centre, The Hague, Netherlands
[4]Department of Surgery, Leiden University Medical Centre, Leiden, Netherlands
[5]Department of Radiology, Maastricht University Medical Centre+, Maastricht, Netherlands
[6]Department of Internal Medicine, Division of Nephrology, Maastricht University Medical Centre+, Maastricht, Netherlands
[7]Department of Health Services Research, CAPHRI Care and Public Health Research Institute, Maastricht University, Maastricht, Netherlands
[8]Department of Internal Medicine, Leiden University Medical Centre, Leiden, Netherlands

**Collaborators** OASIS study group: Marijke J Molegraaf (Department of Vascular Surgery, Isala, Zwolle, the Netherlands); Philippe WM Cuypers (Department of Vascular Surgery, Catharina Ziekenhuis, Eindhoven, the Netherlands). Denise Nio (Department of Vascular Surgery, Spaarne Gasthuis, Haarlem, the Netherlands); Cora H Arts (Department of Vascular Surgery, Medisch Centrum Leeuwarden, Leeuwarden, the Netherlands); Jacobien C Verhave (Department of Internal Medicine, Rijnstate, Arnhem, the Netherlands); Çagdas Ünlü (Department of Vascular Surgery, Noordwest Ziekenhuisgroep, Alkmaar, the Netherlands); Patrick WHE Vriens (Department of Vascular Surgery, Elisabeth-TweeSteden Ziekenhuis, Tilburg, the Netherlands); Goos D Laverman (Department of Internal Medicine, Ziekenhuisgroep Twente, Almelo, the Netherlands); Aron S Bode (Department of Vascular Surgery, Canisius Wilhelmina Ziekenhuis, Nijmegen, the Netherlands); Marcel C Weijmer (Department of Internal Medicine, OLVG, Amsterdam, the Netherlands); Bas Govaert (Department of Vascular Surgery, Máxima Medisch Centrum, Veldhoven, the Netherlands); Aaltje Y Adema (Department of Internal Medicine, Medisch Centrum Leeuwarden, Leeuwarden, the Netherlands); Gijs MJM Welten (Department of Vascular Surgery, Franciscus Gasthuis & Vlietland, Rotterdam, the Netherlands); Michel MPJ Reijnen (Department of Vascular Surgery, Rijnstate, Arnhem, the Netherlands); Marc JH Groeneveld (Department of Internal Medicine, Haaglanden Medisch Centrum, The Hague, the Netherlands); Hans S Brink (Department of Internal Medicine, Medisch Spectrum Twente, Enschede, the Netherlands); Jeroen B van der Net (Department of Internal Medicine, Albert Schweitzer Ziekenhuis, Dordrecht, the Netherlands); Arno Kuijper (Department of Internal Medicine, Máxima Medisch Centrum, Veldhoven, the Netherlands); Hilde PE Peters (Department of Internal Medicine, Isala, Zwolle, the Netherlands); Maarten A Lijkwan (Department of Vascular Surgery, Albert Schweitzer Ziekenhuis, Dordrecht, the Netherlands); Edith M Willigendael (Department of Vascular Surgery, Medisch Spectrum Twente, Enschede, the Netherlands); Roos C van Nieuwenhuizen (Department of Vascular Surgery, OLVG, Amsterdam, the Netherlands); Rombout R Kruse (Department of Vascular Surgery, Ziekenhuisgroep Twente, Almelo, the Netherlands); Yvonne C Schrama (Department of Internal Medicine, Franciscus Gasthuis & Vlietland, Rotterdam, the Netherlands); Karina A de Groot (Department of Internal Medicine, Noordwest Ziekenhuisgroep, Alkmaar, the Netherlands); Karima Farhat (Department of Internal Medicine, Spaarne Gasthuis, Haarlem/Hoofddorp, the Netherlands).

**Contributors** BDCH drafted the manuscript. MGJS designed the study and supervised the writing of the manuscript. CLR and FWD performed the external validation of the mortality prediction score. KEAvdB, MWdH, MHH, MJCH, MMvL, JIR, JHMT and GWHS provided significant intellectual input in the development of the study protocol. All authors read and approved the final manuscript.

**Funding** This study was funded by Leading the Change (ZonMw project number 80-85009-98-2004). The funder has no role in the design and management of the trial, nor in the data collection and analysis or interpretation and publication of the results.

**Competing interests** None declared.

**Patient consent for publication** Not applicable.

**Ethics approval** This study involves human participants and was approved by METC (Medical Ethics Review Committee) azM/UM (Maastricht University Medical Centre+/Maastricht University). NL70385.068.19 / METC19-053Participants gave informed consent to participate in the study before taking part.

**Provenance and peer review** Not commissioned; externally peer reviewed.

**Data availability statement** After completion of the Optimising Access Surgery in Seniors Haemodialysis Patients trial, data will be made available for further research and verification and will therefore be documented and saved in an accessible manner that is in accordance with the laws and regulations concerning privacy sensitive data.

ORCID iDs
Boudewijn DC Heggen http://orcid.org/0000-0001-8978-4128
Chava L Ramspek http://orcid.org/0000-0002-7883-5927
Koen E A van der Bogt http://orcid.org/0000-0002-5712-5723
Mickaël J C Hiligsmann http://orcid.org/0000-0003-4274-9258

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
