## [Reviewer comments · BMJ Open]

ARTICLE DETAILS

TITLE (PROVISIONAL)	Optimising Access Surgery In Senior haemodialysis patients (OASIS): Study protocol for a multicentre randomised controlled trial
AUTHORS	Heggen, Boudewijn; Ramspek, Chava; van der Bogt, Koen; de Haan, Michiel; Hemmeler, Marc; Hiligsmann, Mickaël; van Loon, Magda; Rotmans, Joris; Tordoir, Jan; Dekker, Friedo; Schurink, Geert-Willem; Snoeijs, Maarten

VERSION 1 – REVIEW

REVIEWER	Gonzalez F., Fernando University of Chile, Nephrology
REVIEW RETURNED	27-Jun-2021

GENERAL COMMENTS	Comments on Optimising Access Surgery in Senior haemodialysis patients (OASIS): Study protocol for a multicentre randomized controlled trial The trial rationale is very well thought and focused in a real problem as it is how to choose the most favorable vascular access for elderly patients who require hemodialysis. The authors shortly describe the state of the art in this field from the vascular surgeon as well as the nephrologist perspective. Study aim is to compare vascular access both central catheters and arteriovenous (AV) grafts are better choices than native AV fistulas. Nevertheless, in an elderly population with a high risk of die during the next year after requiring of hemodialysis, a more realistic investigative and clinical question could be which vascular access is more globally convenient to the elderly patient considering the three objective of the trial, as they are a reduction in access-related intervention rates, greater patient satisfaction, and lower health care costs. Methods Inclusion criteria 4: Please, consider replace “Fit for vascular access surgery” by “Fit for any of the proposed vascular access interventions”. Inclusion criteria 5b: Consider to include patients in hemodialysis with just one central catheter in the time window for recruit. More catheters could be associated to more vascular complications making comparisons unfair. What is the standard catheter in use? Tunneled or not?
---

	Inclusion criteria 7: Please, define that only any of the internal jugular veins are to be used for a valid access in the trial (Exclusion criteria: Any other central venous access different from any of both internal jugular veins) Exclusion criteria 4: include patients who attending nephrologists are not sure if they are really candidates to receive hemodialysis at the moment to be recruited. Tunneled catheters: Please define what type of catheter is intended to be used during the trial. Single or multiple lumens? Primary study outcome: As early dying is a competitive risk in the trail, the three types of vascular access will have different follow-up times: 3-6 months for fistulas, 2 weeks for grafts and a couple of days for catheters, please clarify how is going to be manage this issue as well as the asymmetrical times frames from randomization to real use of any of the three vascular accesses. Sample size calculation: Please consider my earlier comment about comparing statistically the three interventional alternatives. Another issue is to consider to better define to end the trial when all the statistically estimated outcome events have already happened and to calculate the effect of the competing risk of early death with the access procedures. References, Figures, Tables and Abstract are all OK.
--	--

REVIEWER	Murea, Mariana Wake Forest Baptist Medical Center
REVIEW RETURNED	05-Jul-2021

GENERAL COMMENTS	Heggen and colleagues report the design of a multicenter clinical trial, conducted in the Netherlands, that randomly allocates adults 70 years of age and older, approaching dialysis or on hemodialysis with a catheter, to a strategy of fistula (usual care group) vs graft vs catheter vascular access. Investigators plan to enroll 195 patients with a follow-up of 1 year from last patient enrolled. This reviewer agrees this is a very important study to be performed, as current clinical practice for vascular access approach is based solely on retrospective / observational studies. Main comments: Abstract:  - Change “an unusually long life expectancy” to a more specific statement - The investigators indicate the patients allocated to usual care will be treated according to current guidelines and undergo AVF creation. However, recent guidelines are no longer AVF-centric. We recommend changing this statement in the abstract Eligibility criteria:  a) Provide literature reference concerning eligibility criteria for the arterial and venous diameter that determine fistula and graft suitability b) Will all the patients with prior unsuccessful arteriovenous fistula or graft vascular access surgery be excluded? Even if they still have vascular anatomy suitable for all types of vascular access? Justify their exclusion if the answer is yes. c) It is reasonable to exclude patients with <6mo expected survival. However, limiting inclusion to those with expected median survival of 1.5-3 years will inherently reduce ‘real world’ generalizability of the results. Exclusion of elderly patients with
---

	expected median survival of 4-5 years needs justification, especially since the investigators hypothesize catheter and/or graft access strategy will be associated with better patient-reported outcomes. Outcomes: a) It can be argued that the primary outcome elected for this trial (rate of access-related interventions required for each patient-year of haemodialysis treatment) might not be most appropriate for the population studied, for several reasons. First, it is unclear how this will be calculated since the investigators defined the primary outcome per patient-year of dialysis treatment. However, the investigators further state the outcome measure specifically includes interventions before dialysis initiation and after dialysis cessation in the occasional patient who stops haemodialysis etc. Second, if pre-dialysis interventions will be included, the incidence of adjuvant interventions will be confounded by the fact that the impetus for access intervention will differ between pre-dialysis and on-dialysis patients. Retrospective studies showed such a difference exists, i.e., AV accesses placed pre-dialysis receive fewer adjuvant interventions compared with AV accesses placed after hemodialysis was started. Third, it is also unclear why vascular access interventions after hemodialysis is stopped will be included. Again, the approach to vascular access care for those who stop hemodialysis because of a successful transplant versus transition to PD can be totally different in clinical practice. Therefore, mixing so many different scenarios within the primary outcome could significantly reduce the main objective of delineating access outcomes independent of other variables—especially given a relatively small sample size with 3 randomization groups. b) Specify what will be included in the secondary outcome of vascular access complications. c) how will ‘cannulation difficulties’ be captured? What will be the definition of cannulation difficulty? Statistical analysis: a) How will the primary outcome be compared? Will the investigators combine graft and catheter groups vs fistula group? Or analyze graft vs fistula and catheter vs fistula? b) Clarify how the cross over “If the assigned vascular access strategy proves not to be feasible, patients will crossover to one of the other treatment groups after discussion of the case with the principal investigator” be analyzed c) Will a competing event analysis for death be included in the primary analysis? d) One year of follow-up is too short; it will not capture the full extent of fistula interventions, which typically occur later than graft interventions after access placement. e) Frequency of questionnaire administration (monthly and every 3 months) seems too high, posing significant respondent burden and fatigability especially considering the population studied f) The sample size might be too small, taking into account 3 randomization groups. Did the investigators account well for an expected high mortality rate in this population, which would be a competing event in outcome analysis? Even with follow-up of one year from last enrolled participant and enrollment period of 2 years, at least 30% of the patients might have <1 year of follow-up Other comments: 1. External validation of a mortality prediction model: In this patient population, it is unlikely that any or most missing data were missing at random, as assumed. Please comment on the potential
--	--

	impact of this assumption on the sensitivity of your prediction model. 2. Investigators quoted elderly patients have a 30% lower risk of complications of central venous catheters, referencing data from Canadian cohort [Reference 4: Poinen et al, Am J Kidney Dis 2019]. Consider noting similar findings in a US cohort [CJASN April 2014, 9 (4) 764-770]. 3. Page 5, correct type 'stable renal function' to 'stable renal function'
--	---

VERSION 1 – AUTHOR RESPONSE

Reviewer: 1

Dr. Fernando Gonzalez F., University of Chile, Hospital Del Salvador Comments to the Author:

Methods

Inclusion criteria 4: Please, consider replace “Fit for vascular access surgery” by “Fit for any of the proposed vascular access interventions”.

We agree with your suggestion and have edited this section

Page 4, Methods and Analysis, Inclusion criteria:

“Fit for vascular access surgery as determined by the local multidisciplinary vascular access team” was changed to “Fit for any of the proposed vascular access interventions as determined by the local multidisciplinary vascular access team”

Inclusion criteria 5b: Consider to include patients in hemodialysis with just one central catheter in the time window for recruit. More catheters could be associated to more vascular complications making comparisons unfair. What is the standard catheter in use? Tunneled or not?

Our goal is to include patients in a “real-world setting” that are expected to start haemodialysis within six months after inclusion or who have already started haemodialysis treatment for six months or less using a central venous catheter. The latter group has therefore received at least one central venous catheter but some may have received more than one catheter. This reflects the current situation where the choice for permanent vascular access strategy often has to be made for patients who already started haemodialysis treatment and therefore received one or more central venous catheters. Therefore, we have not altered this inclusion criterion.

Study sites are allowed to adhere to their usual care and standard materials. However, as stated in the protocol we do expect the study sites to follow current best practice and guideline recommendations which recommend a tunnelled catheter. The protocol also states that we recommend placing a tunnelled catheter just before the start of haemodialysis. Therefore we expect that the vast majority of catheters placed in participants will be tunnelled. For each central venous catheter placed during the trial we register whether the catheter was tunnelled or not.

Page 6, Methods and Analysis, Interventions:

Clarification on the standard catheter to be used: “Because we aim to compare the different surgical strategies for vascular access creation in everyday practice, we will not interfere with surgical protocols at the study sites. However, we expect the study sites to follow current best practice and guideline recommendations.”

Page 7, Methods and Analysis, Interventions:

Recommendation to use tunneled catheters: "We recommend placing a tunneled catheter just before the start of haemodialysis treatment."

Inclusion criteria 7: Please, define that only any of the internal jugular veins are to be used for a valid access in the trial (Exclusion criteria: Any other central venous access different from any of both internal jugular veins)

As with the previous remark, study sites are allowed to adhere to their usual care and protocols. However, we do expect the study sites to follow current best practice and guideline recommendations which recommend the jugular veins as primary site for a central venous catheter. Therefore we expect that the vast majority of catheters will be placed in the Jugular veins. For each central venous catheter placed during the trial we register which vein was used.

Other temporary central venous catheters, for example femoral catheters for acute haemodialysis treatment, are not considered an exclusion criteria.

Page 6, Methods and Analysis, Interventions:

Clarification on the choice of vein to be used: "Because we aim to compare the different surgical strategies for vascular access creation in everyday practice, we will not interfere with surgical protocols at the study sites. However, we expect the study sites to follow current best practice and guideline recommendations."

Exclusion criteria 4: include patients who attending nephrologists are not sure if they are really candidates to receive hemodialysis at the moment to be recruited.

Tunneled catheters: Please define what type of catheter is intended to be used during the trial. Single or multiple lumens?

Current guidelines recommend the referral of patients for vascular access creation, if the attending nephrologist expects that haemodialysis treatment will start within 6 months. It is at this time that the choice for vascular access strategy is usually made. Patients with chronic kidney disease for whom the attending nephrologist is not sure if they are haemodialysis candidates would fall outside of the scope for this trial.

With regards to the standard catheter in use. The OASIS trial is a comparative effectiveness study, which means study sites are allowed to adhere to their usual care and standard materials. For each central venous catheter used in the trial we register if the catheter was single- or dual-lumen.

Page 6, Methods and Analysis, Interventions:

Clarification on the type of catheter to be used: "Because we aim to compare the different surgical strategies for vascular access creation in everyday practice, we will not interfere with surgical protocols at the study sites. However, we expect the study sites to follow current best practice and guideline recommendations."

Primary study outcome: As early dying is a competitive risk in the trail, the three types of vascular access will have different follow-up times: 3-6 months for fistulas, 2 weeks for grafts and a couple of days for catheters, please clarify how is going to be manage this issue as well as the asymmetrical times frames from randomization to real use of any of the three vascular accesses.

The OASIS trial is based on an intention to treat analysis. Since we include and randomise patients at the time of referral for vascular access creation, the participants follow-up time will start at randomisation (choice for vascular access strategy) and will therefore not be affected by the allocated

vascular access strategy. By starting follow-up at randomisation, we avoid a possible selection bias and immortal time bias as seen in studies which start follow-up at the time a vascular access is created. This is considered as one of the strengths of the OASIS trial.

As the placement of the arteriovenous graft and central venous catheter is generally done just before start of haemodialysis treatment, the participants in the catheter and graft group will most likely be observed long before vascular access creation, whereas patients who are allocated to the autologous arteriovenous fistula group will usually undergo fistula creation within weeks after inclusion. This reflects the current real-world situation and is part of the research question.

Page 5, Methods and analysis, study population

We added the following sentence to help clarify the start of follow-up: "Follow-up starts immediately after treatment allocation."

Sample size calculation: Please consider my earlier comment about comparing statistically the three interventional alternatives. Another issue is to consider to better define to end the trial when all the statistically estimated outcome events have already happened and to calculate the effect of the competing risk of early death with the access procedures.

If we would statistically compare all three groups we would need a larger sample size to correct for multiple comparisons. In order to avoid needing a larger sample size we opted to compare the two intervention (i.e. the arteriovenous graft and central venous catheter) groups to the usual care (autologous arteriovenous fistula) group.

We chose a fixed endpoint for the OASIS trial where all study sites can stop the follow-up, rather than a flexible endpoint based on the number of events. This fixed endpoint was necessary in order to create a plan and apply for funding for the trial.

The primary outcome is the number of interventions per dialysis year and not a "time to event" outcome. Interventions and death are events that can both occur in the same participant during follow-up. However, since a substantial proportion of participants is expected to die before contributing one year of follow-up we adopted a variable follow-up with trial closeout for all patients at the time when the last patient enrolled has one year of follow-up. Since participant recruitment is expected to take 2 years, the resulting additional follow-up time will more than compensate for the loss of follow-up due to early patient mortality.

Page 8, Methods and analysis, sample size calculation

Rationale concerning additional follow-up time due to the variable close-out to compensate for early participant mortality: "Since the study includes patients with a limited life expectancy, a substantial proportion of participants is expected to die before contributing one year of follow-up. To compensate for the resulting loss of statistical power, the trial has a variable follow-up with trial closeout for all patients at the time when the last patient enrolled has one year of follow-up. Since participant recruitment is expected to take 2 years, the resulting additional follow-up time will more than compensate for the loss of follow-up due to patient mortality in the first year of the study period."

Page 9, Methods and analysis, data analysis plan

We added additional information to clarify the terms "intervention" and "usual care" group "Both intervention (i.e. the prosthetic arteriovenous graft and central venous catheter) groups will be compared to the usual care (autologous arteriovenous fistula) group"

References, Figures, Tables and Abstract are all OK.

Reviewer: 2

Dr. Mariana Murea, Wake Forest Baptist Medical Center Comments to the Author:

Main comments:

Abstract:

Change “an unusually long life expectancy” to a more specific statement

The “an unusually long life expectancy” is described and defined in the section “external validation of a mortality prediction model”. As only 22% of the population is expected to have a life expectancy of 5.0 years and the large majority of this population is expected to have a life expectancy of 2.2 years we believe that “unusually long” well describes this principle. Given the word count, the abstract unfortunately does not leave room to further elucidate this.

Page 5-6, Methods and analysis, External validation of a mortality prediction model

The following section explicates how we defined an unusually long life expectancy: “Finally, we determined a sensible cut-off point for identifying a subset of patients with an unusually long life expectancy. A cut-off between a score of <4 and ≥ 5 was chosen to separate patients with unusually good outcomes (consisting of 22% of patients in the validation cohort with a median survival of 5.0 years (95% CI: 4.3-5.4)) from elderly haemodialysis patients with a poor life expectancy (consisting of the remaining 78% of patients in the validation cohort with a median survival of 2.2 years (95% CI: 1.9-2.5))”

The investigators indicate the patients allocated to usual care will be treated according to current guidelines and undergo AVF creation. However, recent guidelines are no longer AVF-centric. We recommend changing this statement in the abstract

We agree that recent guidelines are less AVF-centric and have therefore changed the manuscript. The revised 2019 KDOQI guidelines have a more patient centred approach and leave room for the (permanent) use of a prosthetic arteriovenous graft or central venous catheter. However, autologous arteriovenous fistulas in general are still preferred over grafts and catheters in the current ESVS guidelines and therefore largely considered as usual care. In The Netherlands, around 75% of all haemodialysis patients have a autologous arteriovenous fistula. Therefore, we altered this section of the abstract to make it more accurate.

Page 2, Abstract, Introduction

We changed the sentence “the autologous arteriovenous fistulas that are recommended by current guidelines” to the following: “the autologous arteriovenous fistulas that are typically created in usual care”.

Eligibility criteria:

a) Provide literature reference concerning eligibility criteria for the arterial and venous diameter that determine fistula and graft suitability

The suitable vascular access anatomy was derived from the ESVS guidelines, we have added this source to the concerning criterion.

Page 5, Methods and analysis, study population

The 2018 Clinical Practice Guidelines of the European Society for Vascular Surgery by Schmidli et. al. has been added as source for arterial and venous diameter.

b) Will all the patients with prior unsuccessful arteriovenous fistula or graft vascular access surgery be excluded? Even if they still have vascular anatomy suitable for all types of vascular access? Justify their exclusion if the answer is yes.

All patients with prior unsuccessful arteriovenous fistula or graft vascular access surgery will be excluded. Patients with non-maturation of a fistula are a specific subgroup that has previously been associated with worse outcomes than patients with a successfully matured fistula. Furthermore, our aim is to determine the best vascular access strategy for elderly haemodialysis patients that are expected to start haemodialysis within six months or who have already started haemodialysis treatment for six months. Patients after unsuccessful graft or fistula creation would have a delay to inclusion since they first underwent vascular access creation (and possible interventions to promote maturation) instead of being included directly at the time of first referral for permanent vascular access creation.

c) It is reasonable to exclude patients with <6mo expected survival. However, limiting inclusion to those with expected median survival of 1.5-3 years will inherently reduce 'real world' generalizability of the results. Exclusion of elderly patients with expected median survival of 4-5 years needs justification, especially since the investigators hypothesize catheter and/or graft access strategy will be associated with better patient-reported outcomes.

The limitation to include only patients with an expected median survival of 2.2 years, still allows us to include approximately 80% of all elderly patients referred for permanent vascular access creation. There is clinical equipoise in elderly haemodialysis patients because the balance between risks and benefits of the different types of vascular access is different in this population with a limited life expectancy. Any increase in access-related complications associated with long-term use of grafts and catheters may be less relevant for this group of elderly patients with an expected median survival of 2.2 years.

And although functional fistulas are generally associated with the best long-term outcomes, they may experience more short term problems like non-maturation. These short term problems might be of more importance to elderly with a limited life expectancy.

Furthermore, as observational evidence associates catheters with a higher mortality we chose to exclude the small subgroup of patients with an unusually long life expectancy.

Page 3, Introduction

Here we address the issue of the different balance between risks and benefits in elderly:

“Furthermore, the increase in long-term complications associated with arteriovenous grafts may be less relevant for elderly patients with a limited life expectancy.”

Page 5, Methods and analysis, External validation of a mortality prediction model

In this section we also discuss the issue of the different balance between risks and benefits in elderly and substantiate the exclusion of patients with an unusually long life expectancy: “The majority of elderly haemodialysis patients have multiple comorbidities and a relatively short life expectancy making the long-term performance of the vascular access less relevant. However, for the small group of elderly patients with an unusually long life expectancy, autologous arteriovenous fistulas are considered to be the best vascular access given their low complication rates after maturation.”

Page 10, Ethics and dissemination, ethics

We altered the following sentence to emphasize the clinical equipoise especially applies to elderly with a limited life expectancy “there is clinical equipoise in elderly haemodialysis patients because the balance between risks and benefits of the different types of vascular access is different in this population. Any increase in access-related complications associated with long-term use of grafts and catheters may be less relevant for elderly patients with an expected median survival of 2.2 years.”

Outcomes:

a) It can be argued that the primary outcome elected for this trial (rate of access-related interventions required for each patient-year of haemodialysis treatment) might not be most appropriate for the population studied, for several reasons. First, it is unclear how this will be calculated since the investigators defined the primary outcome per patient-year of dialysis treatment. However, the investigators further state the outcome measure specifically includes interventions before dialysis initiation and after dialysis cessation in the occasional patient who stops haemodialysis etc. Second, if pre-dialysis interventions will be included, the incidence of adjuvant interventions will be confounded by the fact that the impetus for access intervention will differ between pre-dialysis and on-dialysis patients. Retrospective studies showed such a difference exists, i.e., AV accesses placed pre-dialysis receive fewer adjuvant interventions compared with AV accesses placed after hemodialysis was started. Third, it is also unclear why vascular access interventions after hemodialysis is stopped will be included. Again, the approach to vascular access care for those who stop hemodialysis because of a successful transplant versus transition to PD can be totally different in clinical practice. Therefore, mixing so many different scenarios within the primary outcome could significantly reduce the main objective of delineating access outcomes independent of other variables—especially given a relatively small sample size with 3 randomization groups.

1. The primary outcome corresponds to the proposed core outcome measure for haemodialysis vascular access by Viecelli and colleagues. By using this standardized outcome, trials will be easier to compare. The start and end date of haemodialysis treatment for all patients is registered and allows us to measure the number of access-related interventions required for each person-year of haemodialysis treatment. To do this, we will divide the total number of interventions during follow-up by the total number of haemodialysis year

2. Vascular access interventions before the start of haemodialysis treatment are registered because they are part of the standard care which this comparative effectiveness trial evaluates.

3. Vascular access related interventions before dialyse initiation and after cessation of haemodialysis treatment are still relevant to patients. Although interventions after cessation of haemodialysis treatment are expected to occur only sporadically, they may still contribute to the overall burden of vascular access interventions associated with a particular type of vascular access.

Page 7, Methods and analysis, primary study outcome

Concerning the primary outcome we added the following: “To analyse this, we will divide the total number of interventions during follow-up by the total number of haemodialysis years.”

Page 7, Methods and analysis, primary study outcome

Concerning interventions before dialysis initiation and after dialysis cessation we added “These vascular access interventions before dialysis initiation and after dialysis cessation are included as they are part of the standard care that this trial evaluates.”

b) Specify what will be included in the secondary outcome of vascular access complications.

The protocol describes that all access-related complications requiring pharmacological, surgical, or endovascular interventions will be registered from randomisation until the end of the study period or death. The number of days admitted to hospital or visiting out-patient clinics for any reason and for vascular access-related reasons per person-year will also be registered.

Page 7, Methods and analysis, secondary study outcomes

The secondary study outcomes are defined in the following section: "Access-related complications requiring pharmacological, surgical, or endovascular interventions will be registered from randomisation until the end of the study period or death. Furthermore, the number of days admitted to hospital or visiting out-patient clinics for any reason and for vascular access-related reasons per person-year will be registered."

c) how will 'cannulation difficulties' be captured? What will be the definition of cannulation difficulty?

Dialysis nurses are asked on a monthly basis if there were any difficulties cannulation the vascular access during the last month. Dialysis nurses are asked if more than one attempt to cannulate was needed, whether it was impossible to cannulate with to needles or if cannulation was painful

Page 8, Methods and analysis, secondary study outcomes

The following was added: "Cannulation difficulties or failures will be recorded with the use of monthly questionnaires asking dialysis nurses whether more than one cannulation attempt was required, whether two-needle cannulation was impossible or if cannulation was painful."

Statistical analysis:

a) How will the primary outcome be compared? Will the investigators combine graft and catheter groups vs fistula group? Or analyze graft vs fistula and catheter vs fistula?

Both intervention groups will be compared to the usual care group, i.e. we will analyse graft vs fistula and catheter vs fistula.

Page 9, Methods and analysis, data analysis plan

We added additional information to clarify the terms "intervention" and "usual care" group "Both intervention (i.e. the prosthetic arteriovenous graft and central venous catheter) groups will be compared to the usual care (autologous arteriovenous fistula) group"

b) Clarify how the cross over "If the assigned vascular access strategy proves not to be feasible, patients will crossover to one of the other treatment groups after discussion of the case with the principal investigator" be analysed

The primary analysis will be on the intention to treat population. Patients that cross over will therefore remain in the allocated group for the primary analysis. Exploratory on-treatment analyses will be performed as well.

Page 9, Methods and analysis, data analysis plan

The above is mentioned in the following section: "The primary analysis will be on the intention to treat population. Exploratory on-treatment analyses will be performed as well."

c) Will a competing event analysis for death be included in the primary analysis?

A competing event analysis for death will not be included. As the primary outcome is the number of interventions per dialysis year and not a "time to event" outcome. Vascular access related interventions and death are events that can both occur in the same participant during follow-up and therefore are not necessarily competing events.

d) One year of follow-up is too short; it will not capture the full extent of fistula interventions, which typically occur later than graft interventions after access placement.

The follow-up for the last patient that will be included is one year. Since the inclusion phase is expected to take around 2 years and all patients will be followed up until the end of the trial, median follow-up time is expected to be well over one year and even three years for the first participants.

Page 8, Methods and analysis, sample size calculation

Rationale concerning additional follow-up time due to the variable close-out is mentioned in the following section: "To compensate for the resulting loss of statistical power, the trial has a variable follow-up with trial closeout for all patients at the time when the last patient enrolled has one year of follow-up. Since participant recruitment is expected to take 2 years, the resulting additional follow-up time will more than compensate for the loss of follow-up due to patient mortality in the first year of the study period."

e) Frequency of questionnaire administration (monthly and every 3 months) seems too high, posing significant respondent burden and fatigability especially considering the population studied.

The frequency of questionnaire administration was determined after consultation with patient representatives. However, to lower their burden, participants that have started haemodialysis treatment receive the questionnaires during these treatments and get assistance from a dialysis nurse.

Page 8, Methods and analysis, secondary study outcomes

We added the following phrase: "To lower their burden, participants that have started haemodialysis treatment will receive the questionnaires during these treatments and get assistance from a nurse."

Pagina 9, Patient and public involvement

This section mentions that the frequency of questionnaire administration was determined after consultation with patient representatives "Patient representatives also advised on the frequency and duration of patient-reported outcome measurements and approved the patient information letter."

f) The sample size might be too small, taking into account 3 randomization groups. Did the investigators account well for an expected high mortality rate in this population, which would be a competing event in outcome analysis? Even with follow-up of one year from last enrolled participant and enrollment period of 2 years, at least 30% of the patients might have <1 year of follow-up.

Since a substantial proportion of participants is expected to die before contributing one year of follow-up, the trial has a variable follow-up with trial closeout for all patients at the time when the last patient enrolled has one year of follow-up. Since participant recruitment is expected to take 2 years, the resulting additional follow-up time will more than compensate for the loss of follow-up due to patient mortality in the first year of the study period.

Page 8, Methods and analysis, sample size calculation

Section concerning additional follow-up time due to the variable close-out to compensate for early participant mortality: "Since the study includes patients with a limited life expectancy, a substantial proportion of participants is expected to die before contributing one year of follow-up. To compensate for the resulting loss of statistical power, the trial has a variable follow-up with trial closeout for all patients at the time when the last patient enrolled has one year of follow-up. Since participant recruitment is expected to take 2 years, the resulting additional follow-up time will more than compensate for the loss of follow-up due to patient mortality in the first year of the study period."

Other comments:

1. External validation of a mortality prediction model: In this patient population, it is unlikely that any or most missing data were missing at random, as assumed. Please comment on the potential impact of this assumption on the sensitivity of your prediction model.

We distinguish between 3 types of missing data: missing completely at random, missing at random and missing not at random.

1. Missing completely at random. It is completely random, but we can also impute.
2. Missing at random. We can impute this data.
3. Missing not at random. Both exclusion of the participant and imputation give a bias.

The missing values in the NECOSAD cohort will be a combination of 1,2 & 3 but that will always remain an assumption, there is no way to determine the type of missing data. We are however convinced that number 2 (missing at random) is the most common in NECOSAD. We therefore assume that we have captured the reason for the missing data in other factors and can make an estimate of those values by means of imputation.

Page 5, Methods and analysis, External validation of a mortality prediction model

The way we handled missing data is described in the following section: "Missing data were assumed to be largely missing at random. Therefore, 10-fold multiple imputation with fully conditional specification was performed using the R package 'mice'. All variables of the mortality prediction score and the outcome were included in the imputation model."

2. Investigators quoted elderly patients have a 30% lower risk of complications of central venous catheters, referencing data from Canadian cohort [Reference 4: Poinen et al, Am J Kidney Dis 2019]. Consider noting similar findings in a US cohort [CJASN April 2014, 9 (4) 764-770].

Additional source was added to the introduction.

Page 3, Introduction

We added the following source: "Risk of Catheter-Related Bloodstream Infection in Elderly Patients on Hemodialysis" by M. Murea et. al.

3. Page 5, correct type 'stable renal function' to 'stable renal function'

This typing error was corrected.

Page 3, Introduction

"Finally, observational studies based on dialysis registries do not include patients with arteriovenous fistulas created before dialysis initiation that were never used because of patient death or stable renal function."

VERSION 2 – REVIEW

REVIEWER	Gonzalez F., Fernando University of Chile, Nephrology
REVIEW RETURNED	01-Dec-2021

GENERAL COMMENTS	All comments were welcome by the authors and they did all the suggested corrections. If not, they properly justified their decisions.
---

REVIEWER	Murea, Mariana
-----------------	----------------

	Wake Forest Baptist Medical Center
REVIEW RETURNED	30-Nov-2021

GENERAL COMMENTS	The authors have done an excellent job in addressing all the suggestions and in explaining the approach to their study. This will be a pivotal study that will offer answers to critical vascular access-related questions in the vulnerable population of older adults with advanced kidney disease. Thank you.
--